# Antifungal Activity of 6-Methylcoumarin against *Valsa mali* and Its Possible Mechanism of Action

**DOI:** 10.3390/jof9010005

**Published:** 2022-12-20

**Authors:** Yun-Ze Chen, Shu-Ren Wang, Tao Li, Guo-Cai Zhang, Jing Yang

**Affiliations:** 1School of Biological Sciences, Guizhou Education University, Wudang District, Guiyang 550018, China; 2Heilongjiang Province Key Laboratory of Forest Protection, School of Forest, Northeast Forestry University, Hexing Road 26, Xiangfang District, Harbin 150040, China; 3College of Forestry, Guizhou University, Huaxi District, Guiyang 550025, China

**Keywords:** coumarins, 6-methylcoumarin, *Valsa mali*, antifungal mechanism, control effects

## Abstract

*Valsa* canker of apple (VCA) caused by *Valsa mali* severely affected apple production in east Asia. With the increase in drug resistance, there is an urgent need for efficient and environmentally friendly antifungal agents. Coumarins have attracted much attention due to their excellent antimicrobial activity against plant pathogens. In this study, the antifungal activity of several coumarins against phytopathogenic fungi was evaluated, and then the antifungal activity of the screened 6-MCM against *V. mali* and its underlying mechanism was further investigated. The results of the in vitro antifungal activity assay showed that some coumarins had significant inhibitory effects on *V. mali*. Notably, 400 mg/L of 6-MCM had the best antifungal activity of 94.6%. Further experiments showed that 6-MCM slowed down the growth of *V. mali* mycelia and the germination of spores in a concentration-dependent manner, with EC_50_ of 185.49 and 54.62 mg/L, respectively. In addition, 6-MCM treatment increased mycelial conductivity, extracellular protein leakage, and MDA content, resulting in damage to the cell membrane. Moreover, 6-MCM significantly reduced the cell wall degrading enzymes secreted by *V. mali*, including EG, PG and PL, thereby limiting its pathogenic capacity. SEM and TEM results showed that 6-MCM treatment had a significant effect on the morphology and ultrastructure of mycelial cells. Inoculation of isolated apple branches found that the application of 6-MCM effectively inhibited the development of VCA and significantly reduced the incidence. All these results suggest that 6-MCM has the potential as a green substitute for VCA control.

## 1. Introduction

*Valsa* canker of apple (VCA), caused by *Valsa mali*, was first discovered in Japan in 1903 and entered China from southeastern Liaoning Province in 1916. It is one of the important reasons for the economic loss of the apple industry in China [1]. The disease is characterized by cankers on tree trunks, branches and scaffolding [2]. It is reported that the pathogen causes a latent infection and survives in apple trees for more than a year before symptoms appear [3,4]. At present, the control of VCA is mainly achieved by spraying effective fungicides, manual removal of cankers, or pruning weak/dead branches. However, the long-term use of traditional chemical pesticides will bring a series of ecological safety problems [5]. Therefore, it is urgent to find an environment-friendly fungicide with high efficiency and low toxicity to reduce the occurrence and development of VCA.

Biopesticides are considered significant alternatives to classic agrochemicals and play a vital role in plant disease control [6]. Currently, a variety of plant-derived compounds have been developed as lead compounds for designing biopesticides [7]. Given their potent inhibitory effects on microorganisms and induction of plant defenses, the use of plant-derived compounds may be a promising plant disease management strategy. Among them, the coumarin compounds identified in plants have antimicrobial activity against various pathogens in vitro [8]. Thus, coumarins have attracted widespread interest recently based on their excellent antimicrobial activity against plant pathogens.

Coumarin compounds are a general term for a class of natural compounds with a benzo α-pyrone nucleus, which are synthesized through the benzoxazine metabolic pathway [9]. They are widely present in the roots, stems and leaves of plants such as Umbelliferae, Rutaceae, Compositae, Leguminosae, Moraceae and Solanaceae, in the form of free state or glycosides, as well as in some microorganisms and animals. According to the substituents and their positions on the ring, coumarin compounds can be divided into simple coumarins, furanocoumarins, pyranocoumarins, dicoumarins and other coumarins [10]. Coumarin compounds are widely used for their antimicrobial, anticancer, anti-inflammatory and antioxidant activities [11,12,13,14]. 4-methoxycoumarin was found to present an antifungal efficacy in a concentration-dependent manner in vivo and could be used to prevent the black potato scurf [15]. Han et al. initially explored the antibacterial mechanism of a novel coumarin antibiotic, 7-methoxycoumarin, and found that 7-methoxycoumarin significantly inhibited the bacterial growth of *Ralstonia solanacearum* and effectively inhibited the incidence of tobacco bacterial wilt [16]. Ali et al. identified a novel natural antifungal compound, 5′-hydroxy-auraptene, a coumarin derivative extracted from *Lotus lalambensis*, which has good antifungal and anti-aflatoxigenic activities against *Aspergillus flavus* [17]. Moreover, 6-methylcoumarin was found to cause cell elongation, disrupt cell division, and inhibit the expression of the protein-coding gene *ftsZ* of *R. solanacearum* [18].

However, the inhibitory effect and mode of action of coumarin compounds on forest diseases are still unclear. Therefore, in the present study, the mycelial growth inhibition rate of five coumarin compounds against six forest pathogenic fungi was used as the screening basis, and the best 6-methylcoumarin (6-MCM) was selected as the plant-derived antifungal agent. Then, the antifungal activity, control effect and possible mechanism of 6-MCM on *V. mali* were explored. This study is aimed to provide a theoretical reference for the development and application of 6-MCM as an alternative new plant-derived antifungal agent to prevent and control VCA, as well as other forest pathogenic fungi.

## 2. Materials and Methods

### 2.1. Chemicals

Coumarins (analytical purity, >98%) were purchased from Macklin Biochemical Technology Co., Ltd. (Shanghai, China), and the chemical structures are shown in Figure 1. Ten percent Tween 80 (*v*/*v*) was used as a solvent to prepare a stock solution of coumarins in different concentrations. Malondialdehyde (MDA) content, superoxide dismutase (SOD), catalase (CAT), polygalacturonase (PG), pectin lyase (PL), and endo-1,4-β-D-glucanase (EG) test kits were sourced from Suzhou Grace Biotechnology Co., Ltd. (Suzhou, China). Analytical grade reagents and solvents were sourced from Fuyu Fine Chemical Co., Ltd. (Tianjin, China).

### 2.2. Pathogens and Cultures

The forest pathogenic fungi *Pestalotiopsis neglecta*, *Valsa mali*, *Botrytis cinerea*, *Fusarium oxysporum*, *Cytospora chrysosperma* and *Sphaeropsis sapinea* were provided by Heilongjiang Province Key Laboratory of Forest Protection (Northeast Forestry University, Harbin, China) and was maintained on potato dextrose agar (PDA) medium at 25 °C. Spores of *V. mali* were harvested by pouring 1% Tween-80 (*v*/*v*) solution into a 7-day-old plate and vortexing for 30 s. The spore suspensions were filtered through sterilized cotton. A hemocytometer under 400× magnification via inversed optical microscopy (Motic B1 Series) was used to adjust spore concentrations to 1 × 10^6^ spores/mL [19].

### 2.3. Antifungal Activity of Coumarins against Forest Pathogenic Fungi

The antifungal activities of coumarins against *P. neglecta*, *V. mali*, *B. cinerea*, *C. chrysosperma*, *F. oxysporum* and *S. sapinea* were determined following the same procedure previously used by our research group [19]. Briefly, mycelial plugs (diameter, 5.0 mm) of 1-week-old fungal cultures were placed at the center of a PDA medium supplemented with 400 mg/L coumarins. Then, the plates were sealed and placed in a 25 °C incubator. Ten percent Tween-80 (*v*/*v*) treatment was used as a negative control. Each treatment was carried out with three biological replicates. After 5 d, mycelial radial growth diameters were measured, and the inhibition rates were then calculated.

### 2.4. Antifungal Activity of 6-MCM against V. mali

The mycelial growth inhibition rate of the 6-MCM screened out was determined against *V. mali* using the method described above, with a concentration gradient of 25, 50, 100, 200, 400 and 800 mg/L.

Furthermore, the hanging drop method was used to assess the effect of 6-MCM on spore germination of *V. mali*, with a concentration gradient of 25, 50, 100, 200, 400, 800 and 1600 mg/L. The final 40 μL of each mixture was separately added dropwise to a hemocytometer and placed in a 25 °C incubator. After 24 h, the number of germinated spores was recorded, and the inhibition rate of spore germination was calculated [20]. Each treatment was conducted in triplicates, and approximately 200 spores were randomly selected for microscopic examination in each replicate.

### 2.5. Effects of 6-MCM on the Cell Membrane of V. mali

The spore suspension of *V. mali* was inoculated into a PDB medium containing 100 mg/L 6-MCM and cultured at 25 ± 1 °C after shaking at 150 rpm. After incubation for 48 h, the hyphae were collected every 12 h until 96 h, using bibulous paper and washed thrice with 0.9% NaCl. Every 0.5 g (fresh weight) was then resuspended in various concentrations of 6-MCM solutions (0, 50, 100 mg/L) and incubated at 25 ± 1 °C. The supernatant was collected at 0, 2, 4 and 6 h and used for extracellular conductivity and cellular leakage determination. A DDS-11 digital conductivity meter (Shanghai Precision Scientific Instrument Co., Ltd., Shanghai, China) was used to measure the extracellular conductivity (μS/cm). Soluble protein leakage was measured following the method described by Bradford [21], and N6000 dual-beam UV/visible spectrophotometry (Shanghai Youke Instrument Co., Ltd., Shanghai, China) was used according to the manufacturer’s instruction to determine soluble sugar leakage. Each treatment was replicated thrice, and the experiments were repeated twice.

The effect of 6-MCM on the membrane lipid peroxidation damage of *V. mali* was detected by measuring malondialdehyde (MDA) content, following instructions. The mycelia treated with 100 mg/L 6-MCM for 48, 60, 72, 84 and 96 h were harvested for evaluation, as earlier described. Each treatment had three replicates, and the experiments were repeated twice.

### 2.6. The Effect of 6-MCM on V. mali Enzyme Activity

The earlier collected mycelia (2.5.1) were used to assess SOD and CAT activities, and the supernatant was used to assess PG, PL, and EG activities. Kits and N6000 dual-beam UV/Visible spectrophotometry (Shanghai Youke Instrument Co., Ltd., Shanghai, China) was used following instructions to determine the enzyme activities. Each treatment was replicated thrice, and the experiments were repeated twice.

### 2.7. Scanning Electron Microscopy (SEM)

The effect of 6-MCM on *V. mali* mycelial morphology during growth was investigated using SEM. Spore suspensions were inoculated in a PDB medium with different 6-MCM concentrations (0, 105 mg/L) and then cultured at 25 ± 1 °C after shaking at 150 rpm. After 3 d, the hyphae (1 g, fresh weight) were collected and treated using the method described by Ji et al. for observation under SEM (JSM-7500F, JEOL, Tokyo, Japan) at 4000× magnification [22].

### 2.8. Transmission Electron Microscopy (TEM)

The effect of 6-MCM on *V. mali* mycelial ultrastructure during growth was investigated using TEM. Hyphae (1 g, fresh weight) was collected as described above, and sample preparation was performed with reference to the method of Yang et al. [20]. The hyphae were fixed in 2% (*v*/*v*) glutaraldehyde solution at room temperature for 4 h and then rinsed 2–3 times with 0.1 M phosphate-buffered saline (PBS, pH 7.2) for 15 min each. Then they were fixed with 0.1 M samarium tetroxide (OsO_4_) for 2 h at 4 °C and dehydrated with 30%, 50%, 70%, 80%, 90% and 100% ethanol gradients, respectively, with a treatment time of 15 min of each concentration. After the samples were embedded for 24 h, the hyphal sections of about 70 nm were cut with an ultramicrotome (EM UC6, LEICA, German) and stained with 2% uranyl acetate and lead citrate. Finally, the samples were observed under TEM (H7650TEM, HITACHI, Tokyo, Japan) at a 25,000× magnification.

### 2.9. Control Effect

Healthy apple branches of similar length are washed with sterile water for surface dust, disinfected with 2% (*v*/*v*) sodium hypochlorite solution, and then air-dried. The bark about 1 cm from the centerline of the apple branch was removed with a peeler. The 7-day-old mycelial plug of *V. mali* (5.0 mm in diameter) was inoculated at the center of the apple branch, then sealed with plastic wrap sprayed with a small amount of sterile water, and placed in a humidity-controlled room with a constant temperature of 25 °C. After 12 h, the plastic wrap was torn off, and the plug on the branches was removed [1]. Then 6-MCM with a concentration of 0.8 and 1.6 mg/mL was sprayed on the apple branches, respectively. An equal amount of 10% Tween-80 (*v*/*v*) was used as a negative control, and Ningnanmycin was used as a positive control. The treated apple branches were placed in a humidity-controlled room at 25 °C for 4 d, and they were taken out again to expand the bark removal area with a peeler to measure the lesions. Three replicates were set for each treatment, with six branches per replicate.

### 2.10. Statistical Analysis

Excel 2019 (Microsoft Inc., Seattle, WA, USA) was used to calculate mean values and standard deviations (n = 3). One-way ANOVA and student’s *t*-test were performed in SPSS Statistics 24.0 (SPSS Inc., Chicago, IL, USA). Statistical significance was set at *p* values < 0.05. Origin Pro 9.1 (OriginLab Inc., Northampton, MA, USA) and Publisher 2019 (Microsoft Inc., Seattle, WA, USA) were used for the graphics.

## 3. Results

### 3.1. Antifungal Activity of Coumarin Compounds against Forest Pathogenic Fungi

The preliminary in vitro antifungal activities of five coumarin compounds at a concentration of 400 mg/L were tested by the mycelial growth rate method. It can be seen from Figure 2 that all five coumarin compounds have certain inhibitory effects on the mycelial growth of *P. neglecta*, *V. mali*, *B. cinerea*, *C. chrysosperma*, *F. oxysporum* and *S. sapinea* at the concentration of 400 mg/L. In particular, 6-MCM had the highest antifungal activity, especially against *V. mali* (94.6%) and *P. neglecta* (76.9%). Thus, the subsequent experiments were conducted on *V. mali*.

### 3.2. Virulence of 6-MCM against V. mali

The virulence of 6-MCM on the mycelial growth and spore germination of *V. mali* is shown in Figure 3. It can be seen from Figure 3A that 6-MCM inhibited the mycelial growth in a concentration-dependent manner. When the concentration was 800 mg/L, the inhibition rate of the mycelial growth reached 100%. After transferring the mycelial plug to a fresh PDA medium without the tested coumarin, *V. mali* still maintained a lack of growth. Similarly, the inhibitory effect of 6-MCM on *V. mali* spore germination was positively correlated with the concentration (*p* < 0.05), and the inhibition rate at 400, 800 and 1600 mg/L was 95.1%, 98.8% and 100%, respectively (Figure 3C).

Probit analysis with SPSS was used to determine the EC_50_ values and the virulence regression equations of 6-MCM against *V. mali* (Table 1). The virulence regression equation against mycelial growth inhibition of *V. mali* was Y = 3.406X − 7.726 (R^2^ = 0.968), with the EC_50_ value 185.49 mg/L, while the virulence regression equation against spore germination inhibition was Y = 2.059X − 3.578 (R^2^ = 0.979), with the EC_50_ value 54.62 mg/L.

### 3.3. Effects of 6-MCM on the Cell Membrane of V. mali

The effect of 6-MCM on *V. mali* cell membrane permeability was evaluated by measuring the conductivity and extracellular soluble protein content of *V. mali* mycelial culture medium, and the result is shown in Figure 4. From 48 to 96 h, both the conductivity of the control group and the 6-MCM treatment group showed a downward trend (Figure 4A). However, the conductivity of the 6-MCM treatment group was always significantly higher than that of the control group (*p* < 0.05). The soluble protein content of *V. mali* remained basically stable with the extension of the culture time, but it is notable that the soluble protein content of *V. mali* treated with 6-MCM was also significantly higher than that of the control group (*p* < 0.05) and almost doubled (Figure 4B). The results indicated that 6-MCM treatment might destroy the permeability of the cell membrane of *V. mali*, causing the intracellular ions and protein to leak out, thereby affecting the mycelial growth and achieving the antifungal effect.

The effect of 6-MCM on *V. mali* cell membrane lipid peroxidation assessed by MDA content is shown in Figure 5. The MDA content of *V. mali* after 6-MCM treatment was always higher than that of the control group, and there were significant differences between the two groups except at 96 h (*p* < 0.05). Among them, the difference was most significant at 60 h, when the MDA content of the 6-MCM treated *V. mali* mycelia was 6.06 nmol/g, about 2.2 times that of the control (2.79 nmol/g). Interestingly, with the extension of culture time, the mycelial MDA content of the 6-MCM-treated group gradually decreased until the difference from the control was not significant at 96 h, indicating that the antioxidant mechanism might be activated to resist external oxidative damage.

### 3.4. Effects of 6-MCM on Enzyme Activity of V. mali

The effects of 105 mg/L 6-MCM on the activity of exocellular enzymes (EG, PG, and PL) and intracellular enzymes (CAT, SOD) in *V. mali* after incubation for 48–96 h are shown in Figure 6. From 48 to 96 h, 6-MCM treatment significantly decreased the enzyme activity of EG more than that of the control (*p* < 0.05). The EG enzyme activity of the two groups reached the lowest value at 84 h, with the control group at 1170.31 U/mg protein and the 6-MCM treatment group at 572.92 U/mg protein. However, the most significant difference appeared at 48 h, with a *p* value of 0.00001 (Figure 6A). The PG activity in the 6-MCM treatment group was also significantly lower than that of the control (*p* < 0.05), showing an “up-down-stable” trend with prolonged incubation (Figure 6B). In addition, as shown in Figure 6C, the extracellular PL activity of the control group gradually increased with the prolongation of the culture time, while that treated with 6-MCM was significantly inhibited (*p* < 0.05).

As for intracellular enzymes, the activity of CAT and SOD showed opposite trends. The CAT enzyme activity of the treatment group was significantly lower than that of the control group at 48, 60 and 96 h (*p* < 0.05), with the CAT activity of the treatment group reaching the lowest point of 20.13 U/mg protein at 96 h (Figure 6D). In contrast, the SOD activity of *V. mali* after 6-MCM treatment was significantly increased and increased by 407% compared with the control group at 48 h. The difference between the two groups then gradually decreased until it recovered to no significant difference at 96 h (Figure 6E).

### 3.5. Effects of 6-MCM on Microscopic Morphology and Ultrastructure of V. mali Hyphae

The effects of 6-MCM (0 and 105 mg/L) on microscopic morphology (Figure 7) and ultrastructure (Figure 8) of *V. mali* hyphae were observed by SEM and TEM, respectively. The SEM results showed that the untreated *V. mali* hyphae had a complete morphological structure, smooth and flat surface, healthy and plump, showing a good growth state (Figure 7A). However, the 6-MCM-treated *V. mali* hyphae lost their original morphological structure (Figure 7B), and the hyphae surface was rough and uneven, accompanied by swelling, increased branches and other symptoms. TEM results showed that untreated *V. mali* hyphae had uniform cell wall and cell membrane structure, with rich cytoplasmic matrix and intact organelles (Figure 8A), while the cytoplasmic membrane of *V. mali* hyphae treated with 6-MCM detached from the cell wall and caused degradation of the cytoplasmic matrix and organelles (Figure 8B).

### 3.6. Control Effects 

The lesion development of VCA is shown in Figure 9A. As a negative control, 10% Tween-80 treated apple tree branches showed the highest incidence of disease (82.5%). While both concentrations of 0.8 and 1.6 mg/mL 6-MCM showed a control effect on VCA, significantly lower than the control group (*p* < 0.05), and the lesions became smaller with the increase of 6-MCM concentration. Moreover, the control effect of 0.8 mg/mL 6-MCM on VCA was similar to that of the positive control Ningnanmycin (Figure 9B).

## 4. Discussion

VCA is a disease with a long latent period that requires long-term prevention and control management. However, the abuse of traditional chemical pesticides often has certain adverse effects on the environment and non-target organisms, and some pesticides have even been recognized that could be accumulated in the human body through the food chain to cause further risks [23,24]. Therefore, it is extremely important to seek effective and safe chemical substitutes for VCA control. Coumarin, as a natural secondary metabolite in plants, has great potential for development in the fields of medicine, industry and agriculture [23,25,26]. Moreover, it is worth noting that coumarin, as well as its derivatives, have been reported as fungicides to inhibit the growth of agroforestry plant pathogens [8,27]. However, the antifungal properties of coumarins against tree diseases and the mechanism of action of 6-MCM against VCA are rarely reported. In this study, by comparing the antifungal activities of five coumarin compounds against 6 phytopathogenic fungi, 6-MCM was selected as an anti-VCA agent to explore its mechanism of action.

According to our results, 6-MCM has strong antifungal activity against *V. mali,* and the inhibitory effects on the other pathogenic fungi are also considerable. Similarly, Yang et al. found that the antibacterial activity of 6-MCM was also outstanding when they screened the coumarin compounds with the best antibacterial effect against *R. solanacearum* [18]. Further experimental results showed that 6-MCM slowed down the growth of *V. mali* mycelia and the germination of spores in a concentration-dependent manner, with EC_50_ of 185.49 and 54.62 mg/L, respectively. In fact, several studies have demonstrated the antimicrobial activity of various coumarins against plant and animal pathogens. For example, a series of 8-substituted coumarin derivatives synthesized by Wei et al. [28] exhibited potent antifungal activity against four phytopathogenic fungi: *Botrytis cinerea*, *Colletotrichum gloeosporioides*, *Fusarium oxysporum*, and *Valsa mali*. Among them, 8-chloro-coumarin and ethyl 8-chloro-coumarin-3-carboxylate showed the strongest inhibition with EC_50_ of 0.085 and 0.078 mmol/L against *V mali*. However, the EC_50_ values we get are still somewhat different from theirs. Different susceptibility of bacterial and fungal growth to 6-MCM may have contributed to the difference in concentration. As for the comparison with 8-chloro coumarin and ethyl 8-chloro-coumarin-3-carboxylate, it may be as described by Yang et al.: the hydroxylation of C-6, C-7 and C-8 could enhance the antimicrobial activity [29].

The plasma membrane is critical for maintaining the homeostatic environment of fungal cells. Studies have found that coumarin can show strong antibacterial activity against *Escherichia coli* by reducing biofilm formation [30], and hydroxycoumarin also inhibits T3SS and biofilm formation to reduce the pathogenicity of *R. solanacearum* [31], which shows that cell membrane is one of the main targets of coumarins. In the present study, the increase in extracellular conductivity, intracellular protein leakage and MDA content confirmed that 6-MCM treatment caused membrane damage by disrupting the membrane integrity of *V. mali* mycelia and inducing membrane lipid peroxidation. Consistent with our results, Wang et al. found that esculetin, a coumarin derivative derived from natural plant products, also caused cell membrane damage in *Phytophthora capsica* [32]. In addition to the above-mentioned effects of destroying the cell membrane, coumarin can also effectively penetrate the cell membrane and directly affect the hyphal cells. Under SEM and TEM, 6-MCM exhibited obvious destructive effects on *V. mali* hyphae, resulting in surface deformities, wrinkles and ruptures, and changes in intracellular structures. This means that 6-MCM has both fungistatic and fungicidal effects against *V. mali*.

The plant cell wall (PCW), of which the main components include pectin and cellulose, is an important barrier protecting plant cells [33], and fungal pathogens have evolved combinations of plant cell wall degrading enzymes (PCWDEs) to deconstruct them [34]. In addition to secreting toxins, the way in which many pathogenic fungi disrupt plant defense mechanisms is that it produces a variety of PCWDEs [35]. Like other phytopathogenic fungi, *V. mali* secretes multiple toxic compounds and PCWDEs throughout the infection to degrade defense barriers and kill plant cells [36,37]. PCWDEs break down PCW, thereby providing assimilable nutrients for pathogen entry and disease development. Meanwhile, the genome-wide analysis also revealed that *V. mali* contains many genes related to plant cell wall degradation and secondary metabolite biosynthesis [38]. Several studies have investigated the role of PCWDEs genes and enzymes (PL, PG, EG) in the virulence of *V. mali* [39,40,41,42,43]. In this experiment, the extracellular EG, PG, and PL enzyme activities of *V. mali* treated with 6-MCM were lower than those of the control, indicating that 6-MCM may reduce pathogenic enzyme secretion to delay VCA infection.

In the process of exploring new methods to control VCA, biological control is also a very popular research field. Liu et al. studied the biocontrol activity of *Bacillus velezensis* D4 against VCA, and both pot and field test results showed that the D4 strain had a strong ability to control VCA, with a disease prevention rate of 50–60% [1]. However, the application of biocontrol strains needs to consider their colonization ability. It is well known that coumarins play important roles in plant defenses as plant antimicrobial agents or immune inducers [44], so 6-MCM can be somehow used as an effective alternative. In the isolated apple branches inoculated with *V. mali*, the control effect of 800 mg/L 6-MCM treatment on VCA was not significantly different from that of the control agent ningnamycin, but 1600 mg/L 6-MCM treatment was better than ningnamycin (Figure 9), equivalent to the control efficacy of the D4 strain. Although 6-MCM has shown potential as an effective plant-derived antifungal agent against VCA, it is unclear whether this effect can be sustained in different forest environments like ningnanmycin, which is needed for further study.

## 5. Conclusions

In conclusion, the screened 6-MCM has strong antifungal activity against *V. mali*. 6-MCM significantly inhibited mycelial growth. Spore germination of *V. mali* changed the cell membrane permeability and induced oxidative damage to the cell membrane. Pathogenicity-related enzymes, including EG, PG and PL, were also significantly inhibited by 6-MCM. Finally, 6-MCM inhibited the development of *Valsa* canker on isolated apple branches and had a good control effect on VCA. This study shows that 6-MCM has potential application value in the future control of plant diseases such as VCA. Therefore, our study provides an environmentally friendly and effective strategy for the research and development of VCA control agents and may be extended to other plant disease control applications in the future.

## Figures and Tables

**Figure 1 jof-09-00005-f001:**
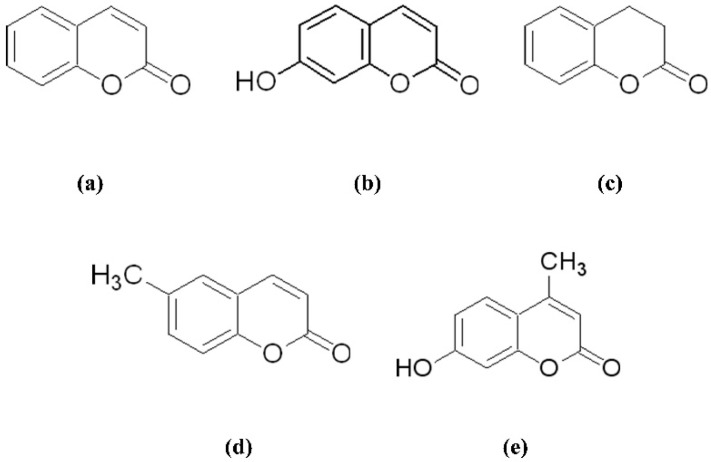
The chemical structures of the studied compounds. Coumarin (**a**), 7-hydroxycoumarin (**b**), dihydrocoumarin (**c**), 6-methylcoumarin (**d**), and 4-methylumbelliferone (**e**).

**Figure 2 jof-09-00005-f002:**
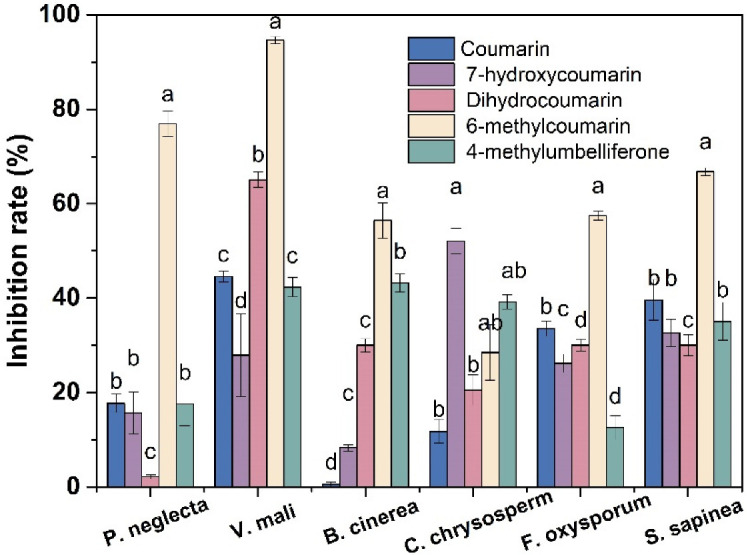
Inhibitory effects of five coumarin compounds on mycelial growth of six phytopathogenic fungi. The bars represent the standard error of the mean (n = 3), and letters (a–d) represent the difference between the inhibitory effects of the five coumarin compounds.

**Figure 3 jof-09-00005-f003:**
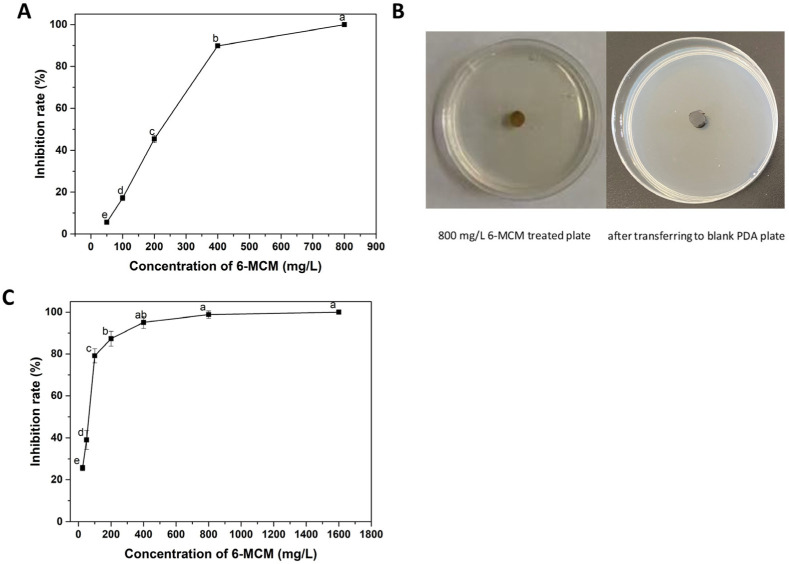
The inhibitory effect of 6-MCM on the mycelial growth (**A**,**B**) and spore germination (**C**) of *V. mali*. The bars represent the standard error of the mean (n = 3), and letters (a–e) represent the difference between the different concentrations of 6-MCM. 6-MCM, 6-methylcoumarin.

**Figure 4 jof-09-00005-f004:**
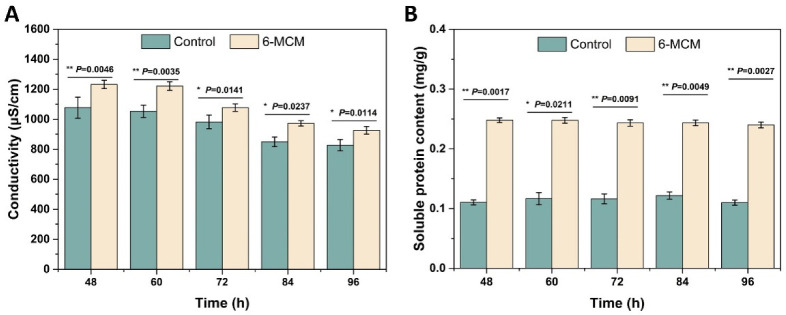
Effects of 6-methylcoumarin on extracellular conductivity (**A**) and soluble protein content (**B**) from *V. mali* mycelia during the incubation for 48 to 96 h. *V. mali* was cultured in the PDB containing 10% Tween-80 (as a control) and 105 mg/L 6-methylcoumarin solution, respectively. The bars represent the standard error of the mean (n = 3), and the asterisks indicate that significant differences exist between 6-MCM treatment and control using an independent-sample *t*-test of 6-MCM. 6-MCM, 6-methylcoumarin.

**Figure 5 jof-09-00005-f005:**
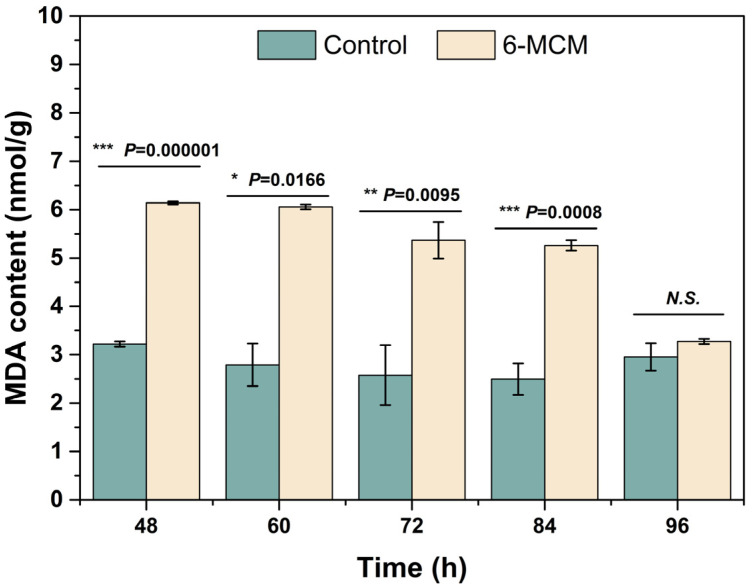
Effect of 6-methylcoumarin on *V. mali* cell membrane lipid peroxidation. *V. mali* was cultured in the PDB containing 10% Tween-80 (as a control) and 105 mg/L 6-methylcoumarin solution, respectively. The bars represent the standard error of the mean (n = 3), and the asterisks indicate that significant differences exist between 6-MCM treatment and control using an independent-sample *t*-test. N.S., no significance. 6-MCM, 6-methylcoumarin.

**Figure 6 jof-09-00005-f006:**
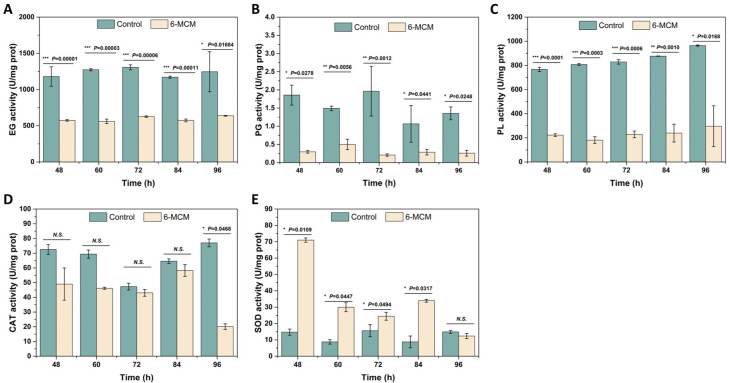
Effect of 6-methylcoumarin on enzyme activity of *V. mali*. (**A**) endo-1,4-β-D-glucanase (EG), (**B**) polygalacturonase (PG), (**C**) pectin lyase (PL), (**D**) catalase (CAT), (**E**) superoxide dismutase (SOD). *V. mali* was cultured in the PDB containing 10% Tween-80 (as a control) and 105 mg/L 6-methylcoumarin solution, respectively. The bars represent the standard error of the mean (n = 3), and the asterisks indicate that significant differences exist between 6-MCM treatment and control using an independent-sample *t*-test. 6-MCM, 6-methylcoumarin. N.S., no significance.

**Figure 7 jof-09-00005-f007:**
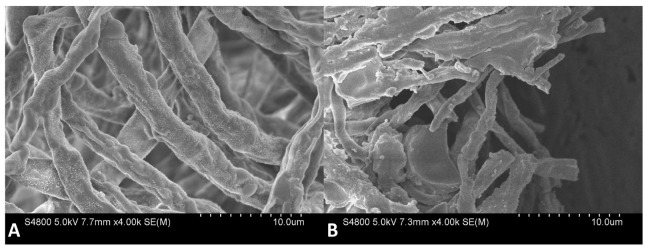
SEM images of *V. mali* mycelial morphology after 6-methylcoumarin treatment: (**A**) Untreated hyphae (normal control) and (**B**) *V. mali* treated with 105 mg/L 6-MCM.

**Figure 8 jof-09-00005-f008:**
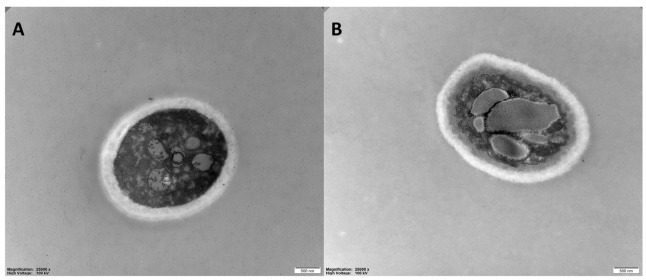
TEM images of *V. mali* mycelial ultrastructure after 6-methylcoumarin treatment: (**A**) Untreated hyphae (normal control) and (**B**) *V. mali* treated with 105 mg/L 6-MCM.

**Figure 9 jof-09-00005-f009:**
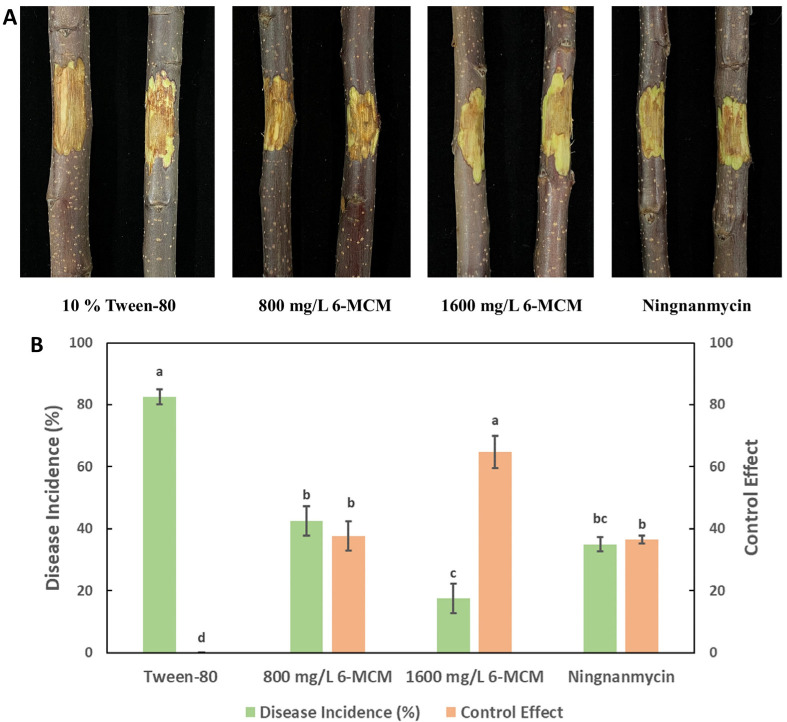
The control effect of 6-methylcoumarin on *Valsa* canker of apple tree branches: (**A**) Lesion symptoms and (**B**) Disease incidence and control effect. The bars represent the standard error of the mean (n = 18), and the letters (a–c) represent the difference between 6-MCM treatment and control of 6-MCM and 6-methylcoumarin.

**Table 1 jof-09-00005-t001:** Virulence regression equation of 6-MCM against *V. mali*.

	Virulence Regression Equation	χ^2^	R^2^	EC_50_ (mg/L)	95% Confidence Interval	EC_90_ (mg/L)	95% Confidence Interval
Mycelial growth inhibition	Y = 3.406X − 7.726	8.995	0.968	185.49	136.849–253.643	441.16	309.576–878.967
Spore germination inhibition	Y = 2.059X − 3.578	6.812	0.979	54.62	46.288–189.456	228.892	189.456–289.685

## Data Availability

Not applicable.

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
