# Peer review of "Antifungal Activity of 6-Methylcoumarin against Valsa mali and Its Possible Mechanism of Action"

_jof, 2022, doi:10.3390/jof9010005_

Round 1

Reviewer 1 Report

Article title:

Antifungal activity of 6-methylcoumarin against Valsa mali and its possible mechanism of action"

I suggest that this article be published in light of research in which the best 6-methylcoumarin (6-MCM) was chosen as the plant-derived antifungal drug after the mycelial growth inhibition rates of five coumarin compounds against six forest pathogenic fungi were used as the screening basis. Next, the 6-MCM's antifungal activity, control impact, and potential mechanism were investigated in relation to V. mali.

The work done is certainly of international interest and the format applied is certainly suitable for a research article. The work is original, of particular interest, and can certainly stimulate research on this topic. The length of the article is appropriate for the journal and the graphs are clear and easy to understand. The conclusion summarizes the aims of the work and future prospects. Just be sure to:

-          Line 49: Change in vitro word to in vitro

-          Line 93: 1×106 spores/mL changed to 1×106 spores/mL

-          Transfer Figure 1 to the Material and Methods section

-          The Figures shown in the results section should be maximized i.e. 3,4,6…etc.

Author Response

Dear Reviewer,

We are very grateful to you for giving me the opportunity to modify the manuscript. Thank you for the comments, the point-by-point responses were listed as follows.

Comment 1: Line 49: Change in vitro word to in vitro.

Response 1: Thanks to your kind reminder, and we have changed in vitro to in vitro in Line 49.

Comment 2: Line 93: 1×106 spores/mL changed to 1×106 spores/mL.

Response 2: Thanks to your kind reminder, and we have changed 1×106 spores/mL to 1×106 spores/mL in Line 98.

Comment 3: Transfer Figure 1 to the Material and Methods section.

Response 3: Thanks to your kind reminder, and we have transferred Figure 1 to Section 2.1 in the Material and Methods.

Comment 4: The Figures shown in the results section should be maximized i.e. 3,4,6…etc.

Response 4: Thanks to your kind reminder, and we have maximized the Figures shown in the results section i.e. 3, 4, 6, 7, 8, 9.

Reviewer 2 Report

Yun-Ze CHEN et al describe “Antifungal activity of 6-methylcoumarin against Valsa mali and its possible mechanism of action”.

Several studies have reported the antifungal activities of coumarin and its derivatives against Apple phytopathogens. So, this work is informative and there is no particular innovation in methodology. In the larger picture, the novelty is limited. However, I recommend accepting it in Journal of Fungi after major corrections.

Some comments:

1.        The author needs to confirm and clarify the novelty of this research. Is the novelty of this research only to evaluate the antifungal properties and the mechanism of action of 6-MCM against VCA or maybe there are other aspects that can be the novelty of this research?

2.        The authors must mention is these apples are treated with chemicals before being treated with coumarins.

3.        The author needs to provide or add photos illustrating in vitro antifungal activities of five coumarin compounds against six forest pathogenic fungi.

4.        The author must justify the choice of the concentration range used (25, 50, 100, 200, 400, 800, 1600 mg/L)?

5.        The authors need to clarify the selection of 10 % Tween-80 (v/v) as a negative control.

6.        The author must provide or add information on the fungitoxic character (fungistatic or fungicidal) of 6-MCM at CMI= 800 mg/L, after transferring the mycelial disc to fresh PDA medium without the tested coumarin.

7.        Lines 178. The coumarins tested were purchased, they are neither synthesized nor extracted from plants, and the sentence” plant-derived” should be deleted.

8.        The authors need to clarify the selection of Ningnanmycin as a positive control for evaluation of Control effect et not also for evaluation of inhibition of mycelial growth.

9.        As for the in vitro antifungal activities of five coumarins, the author should provide more information demonstrating the efficacy of 6-MCM compared to the other coumarins tested. In this part, I am not entirely convinced because hydroxylated coumarins can be more effective. Similarly, the author pointed out that Yang et al., have reported that the hydroxylation of C-6, C-7 and C-8 could enhance the antimicrobial activity (line 337).

10.   The antifungal activity of the compounds tested is essentially due to their interactions with the active sites of the enzymes responsible for the synthesis of the cell wall of the microorganisms (EG, PG, PL, CAT and SOD). Theoretically, molecular docking and molecular dynamics have been performed to explore, predict, visualize and understand protein/enzyme interactions with molecules that could bind specifically to protein active sites. Therefore, the authors are encouraged to explore these computational studies to correlate the antifungal activity of the examined coumarins and their chemical structures and to conclude whether the results obtained are in good agreement with those of the experiment.

Author Response

Dear Reviewer,

I am very grateful to you for giving me the opportunity to modify the manuscript. Thank you for the comments, and we are very appreciative of the valuable comments from you. Based on these comments and suggestions, we have made careful modifications to the original manuscript. All modifications in the revised manuscript were marked in red and highlighted in yellow. The point-by-point responses to your comments are listed in the attached file. We appreciate your warm work earnestly and hope these will make it more acceptable for publication. If there are further issues to be clarified, please contact us without hesitation. Thank you very much again.

Round 2

Reviewer 2 Report

The revised manuscript with the title " Antifungal activity of 6-methylcoumarin against Valsa mali and its possible mechanism of action " (jof-2077843 R1) has been revised in accordance with comments. However, the author should consider these two comments:

Comment 5: The author prepared a stock solution of 6-MCM in 10% Tween 80 (v/v). Afterward, he prepared dilutions to have the desired concentrations. So, it becomes essential to recall this in the experimental procedure.

Comment 6: The author must provide or add this information in Result section (fungicidal effect Figure).

Author Response

Dear Reviewer,

I am very grateful to you for giving me the opportunity to modify the manuscript. Thanks for your kind reminder, and we are very appreciative of the valuable comments from you. Based on these comments and suggestions, we have made careful modifications to the manuscript-R1. All modifications in the revised manuscript were marked in red and highlighted in yellow. The point-by-point responses to your comments are listed as follows. We appreciate your warm work earnestly and hope these will make it more acceptable for publication. If there are further issues to be clarified, please contact us without hesitation. Thank you very much again.

Comment 5: The author prepared a stock solution of 6-MCM in 10% Tween 80 (v/v). Afterward, he prepared dilutions to have the desired concentrations. So, it becomes essential to recall this in the experimental procedure.

Response 5: Thanks for your kind reminder. We have added a sentence of “10% Tween 80 (v/v) was used as solvent to prepare stock solution of coumarins in different concentrations” in Line 82-83. Meanwhile, “control” was changed into “negative control” in Line 107.

Comment 6: The author must provide or add this information in Result section (fungicidal effect Figure).

Response 6: Thanks for your kind reminder. We have added the information in Figure 3 and in-text description (Line 202-203) is added as follows: After transferring the mycelial plug to a fresh PDA medium without the tested coumarin, V. mali still maintained ungrowth.
